# Evaluation of Chemical Elements, Lipid Profiles, Nutritional Indices and Health Risk Assessment of European Eel (*Anguilla anguilla* L.)

**DOI:** 10.3390/ijerph20032257

**Published:** 2023-01-27

**Authors:** Joanna Łuczyńska, Joanna Nowosad, Marek Jan Łuczyński, Dariusz Kucharczyk

**Affiliations:** 1Department of Commodity and Food Analysis, University of Warmia and Mazury in Olsztyn, ul. Plac Cieszyński 1, 10-726 Olsztyn, Poland; 2Department of Ichthyology, Hydrobiology and Ecology of Waters, The Stanisław Sakowicz Inland Fisheries Institute in Olsztyn, ul. M. Oczapowskiego 10, 10-719 Olsztyn, Poland; 3Department of Research and Development, ChemProf, 11-041 Olsztyn, Poland

**Keywords:** mineral component, heavy metals, THQ, MPI, quality indices, benefit–risk ratio, ratio FA:Hg

## Abstract

The concentrations of ten elements (K, Na, Ca, Mg, Zn, Fe, Hg, Cu, Mn, and Cd) and fatty acids were analyzed in muscles of the European eel (*Anguilla anguilla* Linnaeus, 1758). The eels were caught in freshwater lakes connected with the Sawica River (north-eastern Poland). On this basis, it was determined whether the consumption of the fish is beneficial and safe for the health of the consumer. The results showed that the metal concentrations followed this order: K > Na > Ca > Mg > Zn > Fe > Hg > Cu > Mn > Cd. The fatty acids gave rise to the following sequence: MUFAs > SFAs > n-3 PUFAs > n-6 PUFAs. The target hazard quotient (THQ) value was below 1.0. The hazard quotient for the benefit–risk ratio HQ_EFA_ (0.39) also was below one, indicating that the intake of the recommended dose of EPA + DHA (250 mg/day) and the intake of mercury (0.415 mg/kg) for a person weighing 70 kg does not pose an obvious risk for human health. The lipid quality indices were OFA: 24.69, DFA: 74.36, AI: 0.55, and TI: 0.41. Based on the above statements, the consumption of eel meat is safe from a health point of view. However, the levels of toxic metals in the muscles of eels and their environment should continue to be monitored, as eels occupy a high position in the food chain.

## 1. Introduction

The European eel (*Anguilla anguilla* L.) is a species native to the waters of Western Europe [1]. It is a predatory fish [2] and is classified as a catadromous species [2,3,4,5], with reproduction taking place in the Sargasso Sea (Atlantic Ocean). The discovery of small (<10 mm) larvae in the Sargasso Sea by biologist Johannes Schmidt over 100 years ago confirmed the location of the spawning grounds for this species [1]. In Poland, the natural habitat of European eels covers a wide range of water types, with the least significant being rivers and the most important being transitional waters in the Szczecin Lagoon, Vistula Lagoon, and lakes, which are located in the northern part of Poland [2]. 

In addition to proteins and fats, fish require inorganic elements and minerals to function properly [6]. There are various nutritional differences between fish and livestock. Fish require certain lipids that warm-blooded animals do not need, such as n-3 fatty acids for certain species and sterols for crustaceans. Fish also have a unique ability to absorb soluble minerals from the water, reducing the dietary need for certain minerals. Additionally, fish require less energy than warm-blooded animals, resulting in a higher dietary protein/energy ratio for fish. Fish also have a limited ability to synthesize ascorbic acid and must depend on food sources for this essential nutrient [7].

Fish and fish products play an important role in human nutrition, being a source of biologically valuable proteins, fats, and fat-soluble vitamins [8]. Solgi and Beigzadeh-Shahraki [9] observed that fish is one of the best sources of protein and the consumption of fish provides polyunsaturated fatty acids (PUFA), liposoluble vitamins, and essential minerals for human health. Fish is a desirable component of the diet due to its high nutritional value and sensory value [10]. The texture of the flesh and the composition of fat and protein are usually among the major factors determining consumer acceptance. The lipid content and composition are also particularly important in terms of both taste and nutritional value [11]. According to Garcia-Gallego and Akharbach [11], the current intensive techniques for producing European eels have resulted in a product with a high lipid content. Fatty fish contain about 3–20% lipids [12], but some literature data indicate that the fat content of the eel is 19–41% [13]. Belitz et al. [8] noted that eels contain fat at the level of 26% of the edible portion, while protein is 15% of the edible portion. The fat (oil) content of fish varies greatly. This is influenced not only by the species of fish, but also by maturity, season, food availability, and eating habits [8]. Quintaes and Diez-Garcia [14] stated that human nutritional needs require at least 23 minerals. The metals (e.g., Fe, Zn, Cu, and Mn) for which the concentration in the body varies between 0.00001% and 0.01% belong to the group "trace elements". The metals Fe, Zn, Cu, and Mn are also classified as heavy metals [15,16]. The group of heavy metals also includes chromium, cadmium, arsenic, lead, nickel, mercury, and selenium [17,18]. Aquatic ecosystems, both natural reservoirs and aquaculture systems, are contaminated with heavy metals [19]. According to Yousif et al. [20], Agbugui and Abe [17], and Afzaal et al. [21], heavy metals come from anthropogenic sources related to human activity (namely, industrial, agricultural and household waste, and by-products), as well as the combustion of fossil fuels and gasoline, and mining [22]. Garai et al. [22] also mention natural sources of heavy metals, such as volcanic eruption and the weathering of metal-bearing rocks. 

Due to their complex life cycle, catadromous fish species are subject to the cumulative impact of many anthropogenic threats that have led to global decline since the early 20th century [5]. Important factors determining the quality status of the eel are environmental parameters, such as the area of the body of water, depth, connections with the sea, and the size of the river basin [23], while the aspects that explain the storage and elimination of contaminants from the body of these fish are habitat, physiological factors, lipid content, geographic origin, and nutritional behavior [24]. Among these threats, aquatic organisms have a high capacity to absorb heavy metals, including Zn, Cu, Mn, Fe, Cd, and Hg. The first four metals belong to the group of microelements, but when the optimal concentration is exceeded, they can be toxic. These metals are the substances that represent the most dangerous aspects of chemical water pollution, due to their bioaccumulation via the gills, digestive tract, and body surface [22,24], their biomagnification potential, and the fact that they cannot be removed from the body by metabolic activity [25]. Since fish cannot metabolize the metals, even if they do not exceed toxic concentrations, they damage fish growth, development, reproduction, nutrition, and survival ability, affecting physiological, biochemical, metabolic, systemic, and genetic functions [25]. They are non-biodegradable, so they can enter the human body through the consumption of fish and put consumers at risk [19]. Although consumption of fish should be encouraged, it should be noted that consuming fish from a contaminated area poses a risk to human health due to contamination with heavy metals. As a top predator in the food chain, the European eel absorbs many minerals and heavy metals; therefore, this species is often treated as a bioindicator of the quality of aquatic ecosystems [2]. 

Fish, especially fatty fish, contain important n-3 polyunsaturated fatty acids. Fatty acids with five and six double bonds such as eicosapentaenoic acid (20:5n-3; EPA) and docosahexaenoic acid (22:6n-3; DHA) are vital nutrients for humans [13]. Their primary source is marine and freshwater algae, and fish obtain the two fatty acids through the food chain [7]. Chung et al. [26] reported that current dietary guidelines recommend a diet low in saturated fatty acids (SFA) and moderate amounts of monounsaturated fatty acids (MUFA) and polyunsaturated fatty acids (PUFA), including the n-6 and n-3 families, whereas the recommended ratio of n-6 to n-3 typically ranges from 4:1 (or less) to 10:1. According to Valfre et al. [27], the fatty acid profile of the diet plays a significant role in determining its overall health implications. This is particularly true for the relationship between saturated fatty acids (SFA), monounsaturated fatty acids (MUFA), and long-chain polyunsaturated fatty acids (PUFA) [27]. One of nutritional quality indices is considered to be PUFA/SFA. This may be related to the fact that all PUFAs in the diet can lower LDL-C, i.e., low-density lipoprotein cholesterol and lower serum cholesterol, while all SFAs contribute to high serum cholesterol. Thus, the higher the PUFA/SFA ratio, the more positive the effect [28]. These indicators also include AI, which is calculated by adding up the concentrations of the main saturated fatty acids, including 12:0, 14:0, and 16:0, excluding 18:0 (12:0, 16:0, and 4 × 14:0), and dividing them by the total concentration of unsaturated fatty acids (UFA) [29,30], i.e., the ratio of proatherogenic to antiatherosclerotic fatty acids. According to Ulbricht and Southgate [30] and Chen et al. [31], the inverse of the atherogenic index is the HPI, which focuses on the effect of fatty acids composition on CVD. The relationship between the prothrombogenic FAs (C12:0, C14:0, and C16:0) and the antithrombogenic FAs (MUFAs and the n-3 and n-6 PUFAs) is defined as the thrombogenicity index (TI) [28,30]. 

However, it should be remembered that eels are a rich source of protein, vitamins, minerals, and fatty acids recommended in the human diet. On the one hand, consumption of fish should be encouraged. On the other hand, due to contamination with heavy metals, it should be taken into account that consuming fish from a contaminated area poses a risk to human health.

The aims of this study were:To determine the chemical elements, including heavy metals, and the effect of biometric parameters (body weight and total length) on the content of the chemical elements in muscles of the European eel (*Anguilla anguilla* Linnaeus, 1758);To determine, based on quality indicators (THQ, HI, EDI, EWI, HQ_EFA_, and BRQ), whether the level of heavy metal pollution is a concern for the health of the consumer after consumption of the studied fish species;To determine the load of all heavy metals (MPI) in the muscle tissue;To determine, based on the profile of fatty acids and lipid quality indices (PUFA/SFA, OFA, DFA, AI, TI, FLQ, HH, NVI, HPI, OFA, and *de minimis* EPA + DHA:Hg), whether the consumption of the examined fish is beneficial for human health.

## 2. Materials and Methods 

### 2.1. Sampling and Sample Preparation

Migrating female European eels (*Anguilla anguilla* Linnaeus, 1758) (n = 19) were caught in freshwater lakes connected with the Sawica River (in the north-east of Poland). The fish were transported to the laboratory in plastic bags with oxygen. After being anesthetized (MS-222; Finquel, Los Angeles, CA, USA; dose 0.3 mg/L), the fish were marked individually using PIT tags (Biomark, Boise, ID, USA). They were then weighed (±0.1 g) and measured (±0.1 cm) after being brought to the laboratory. The average body weight was 1201.3 ± 248.4 g (719.5–1725.5 g) and the total body length was 86.6 ± 5.8 cm (75.0–97.5 cm). Muscle tissue (without skin) from the dorsal part was harvested after euthanasia. Samples of muscle were prepared separately from each specimen. The samples were ground, homogenized, and stored until analysis in the refrigerator at −30 °C. A plastic knife and fork were used to prepare each specimen. These activities were performed on disposable plastic plates.

### 2.2. Element Analysis

#### 2.2.1. Mercury

The muscle tissue of each individual eel was weighed in two parallel replicates (<270 mg ± 0.0001 g). For this purpose, quartz boats were used. The total mercury was determined using Milestone DMA-80 with dual-cell and UV enhanced photodiodes (Milestone, Sorisole (BG), Italy). The detection limit (LOD) was 0.02 μg/kg. Conditions are presented in an earlier publication [32]. 

#### 2.2.2. Copper, Zinc, Manganese, Iron, Cadmium, Magnesium, Calcium, Sodium, and Potassium

Approximately 10 g samples of muscle (±0.0001 g) in duplicate were dried to constant weight at 105 °C. Then, the samples were combusted at 480 °C for 12 h using laboratory furnaces Nabertherm P330) (Nabertherm GmbH, Lilienthal, Germany). The white ash was dissolved in 1 M HNO_3_ (Suprapur-Merck, Darmstadt, Germany). Each sample was then quantitatively transferred using deionized water (Merck-Millipore Elix Advantage 3, Burlington, MA, USA) into a volumetric flask of volume 25 mL. 

The contents of these metals, except for cadmium, sodium, and potassium, were determined using atomic absorption spectrometry (iCE 3500 Series AAS, Thermo Scientific, Waltham, MA, USA), using cathode lamps appropriate for the given elements and background correction (a deuterium lamp). The absorption technique (acetylene-air flame) was applied to determine the content of Zn, Cu, Fe, Mn, Ca, and Mg. When determining calcium to eliminate the influence of phosphorus, the solution of lanthanum chloride was added (0.5%La^+3^) to all samples and standards. The absorption wavelengths were as follows: 248.3 nm for iron, 213.9 nm for zinc, 324.8 nm for copper, 279.5 nm for manganese, 285.2 nm for magnesium, and 422.7 nm for calcium. The detection limits (LOD) were 0.125 mg/kg for Fe, 0.1 mg/kg for Zn, 0.05 mg/kg for Cu, 0.05 mg/kg for Mn, 0.025 mg/kg for Mg, and 0.5 mg/kg for Ca.

The flame emission technique (acetylene-air flame) was used to determine sodium and potassium content at 589.0 nm and 766.5 nm, respectively. Atomic absorption spectrometry (iCE 3500 Series AAS, Thermo Scientific, Waltham, MA, USA) was used for this purpose. The detection limits (LOD) were 1 mg/kg for Na and 5 mg/kg for K.

The concentration of cadmium was measured using flameless atomic absorption spectrometry FAAS (Thermo Scientific iCE 3500, Waltham, MA, USA) with ZEEMAN background correction and atomization in a graphite cuvette. The cadmium determination was carried out under the following conditions: absorption wavelength for cadmium—228.8 nm; lamp current—50%; slit—0.5 nm; sample volume—20 μL; modifier—Mg(NO_3_)_2_; dry—100 °C; ash—600 °C; and atomize—1000 °C. The detection limit (LOD) was 0.00007 mg/kg. 

Four blanks and four standards were analyzed with each batch of samples. Calibration curves were prepared using four standard solutions (1000 μg/L) with 0.1 M HNO₃ (J.T.Baker^®^, Netherlands). The calibration curves were linear within the range of metal concentration (regression coefficients R^2^ ≥ 0.999).

The quality control of methods was tested using the reference material BCR CRM 422 (lyophilized muscle tissue of cod, *Gadus morhua* (L.)) with a certified value of mercury. The recovery rates were 105.0% Zn, 103.0% Cu, 96% Fe, 103% Mn, 102.9% Cd, and 100.2% Hg, respectively (n = 4).

### 2.3. Fat and Fatty Acids Analysis

Approximately 1 g of each sample in triplicate (0.0001 g) was weighed on a filter and dried to a constant weight at 105 °C. After drying, the samples were transferred to glass thimbles, which were placed in previously weighed beakers. The fat from the fish muscles (without skin) was hot-extracted in three steps (extraction, rinsing, and drying) using an E-816HE automatic extractor (BUCHI, Flawil, Switzerland). After the extraction, all of the petroleum ether was collected in the tank. Fat was dried in a beaker at 100 °C to a constant weight and then weighed. The content of fat (%) was calculated according to the following formula: x = [(b − a) × 100]/c, where: a = weight of flask (g), b = weight of flask with extracted fat (g), and c = weight of samples (g).

The Folch procedure was used to extract fat from the samples [33]. The fatty acid methyl esters were prepared from total lipids with the Peisker method with chloroform: methanol: sulphuric acid (100:100:1 *v*/*v*) [34].

The separation and determination of fatty acids were carried out using the chromatograph 7890A Agilent Technologies (Agilent Technologies, INC., Santa Clara, CA, USA) under the following conditions:Detector: flame ionization (FID);Capillary column (dimension 30 m × 0.25 μm with a 0.32 mm internal diameter, liquid phase StabilwaxR);Temperature:

Flame ionization detector—250 °C; 

Injector—230 °C; 

Column—195 °C;

Carrier gas—helium with a flow rate 1.5 mL/min.

Identification of fatty acids was performed by comparing the retention times of Supelco’s standards (mixture of 37 acids) with the peaks in the test sample. 

### 2.4. Estimated Daily and Weekly Intake (EDI and EWI)

EDI is the estimated daily intake (μg/kg body weight/day)
EDI = C × IR/BW(1)
where C is the average concentration of metals in foodstuffs (μg/g wet weight); IR is the daily ingestion rate (g/daily); and BW is the average body weight (70 kg).
EWI = 7 × EDI(2)

### 2.5. Target Hazard Quotient (THQ)

According to Ahmed et al. [35] and US EPA [36], the THQ estimated the non-carcinogenic health risk to consumers due to the intake of heavy metal contaminated fish using the oral reference dose (RfD) of Cu, Zn, Mn, Fe, Hg, and Cd: 4.00 × 10^−2^, 3.00 × 10^−1^, 1.4 × 10^−1^, 7.00 × 10^−1^, 3.00 × 10^−4^, and 1.00 × 10^−4^, respectively. THQ < 1 means that there are health benefits to consuming this species of fish and there is no risk to the health of the consumer.
THQ = (Efr × ED × FiR × C/RfD × BW × TA) ×10^−3^(3)
where Efr is the exposure frequency (365 days/year); ED is the exposure duration (70 years); FiR is the fish ingestion rate (34.52 g/person/day); C is the average concentration of metals in foodstuffs (μg/g wet weight); RfD is the oral reference dose (mg/kg/day) [36]; BW is the average body weight of local residents (70 kg); and TA is the average exposure time (365 days/year × ED).

### 2.6. Hazardous Index (HI)

HI was calculated from the sum of the hazard quotients of all the metals [35].
HI = ƩTHQ = THQ(Cu) + THQ(Zn) + (THQ(Mn) + THQ(Fe)+ THQ(Hg) + THQ(Cd) (4)

### 2.7. Metal Pollution Index (MPI)

The MPI was calculated using the formulae by Usero et al. [37,38] and Abdel-Khalek et al. [39].
MPI = (M1 ×M2 × M3 × … Mn)^1/n^(5)
where M1 is the concentration of the first metal; M2 is the concentration of the second metal; M3 is the concentration of the third metal; and Mn is the concentration of “n” metal (mg kg^−1^ wet weight) in the muscle tissue.

### 2.8. The Lipid Quality Indices

AI and TI were calculated using the following pattern by Ulbricht and Southgate [30], Garaffo et al. [40], and Telahigue et al. [41].

#### 2.8.1. Index of Atherogenicity (AI)

AI = [C12:0 + (4 × C14:0) + C16:0]/(n-3PUFA+ n-6PUFA + MUFA)(6)

#### 2.8.2. Index of Thrombogenicity (TI)


TI = [C14:0 + C16:0 + C18:0]/[(0.5 × C18:1) + (0.5 × sum of other MUFA) + (0.5 × n-6PUFA) + (3 × n-3PUFA) + n-3PUFA/n-6PUFA)](7)


#### 2.8.3. Flesh-Lipid Quality (FLQ)

The FLQ was calculated in accordance with Abrami et al. [42] and Senso et al. [29].
FLQ = 100 × [EPA + DHA]/[% of total fatty acids] (8)

#### 2.8.4. Hypercholesterolaemic Fatty Acids (OFA)

The calculation formula of OFA is: OFA = C12:0 + C14:0 + C16:0(9)

#### 2.8.5. Hypocholesterolaemic Fatty Acids (DFA)

The DFA was calculated using the formula:DFA = C18:0 + UFA(10)

#### 2.8.6. The Hazard Quotient for the Benefit–Risk Ratio (HQ_EFA_ and BRQ)

Calculation of the hazard quotient for the benefit–risk ratio (HQ_EFA_) were made according to Gladyshev et al. [43].
FP = R_EFA_/C DM = FP × c        HQ = DM/RfD × AW                             HQ_EFA_ = (R_EFA_/C) × c × (1/(RfD × AW)) = (R_EFA_ × c)/(C × RfD × AW)(11)
where EFA is the EPA + DHA (250 mg per day for a human person is presented here as R_EFA_); C is the content of EFA (mg/g); consuming FP, a person will obtain a dose of a metal, DM (µg per day); c is the content of metal in fish (µg/g); RfD is the Reference Dose of Cu, Zn, Mn, Fe, Hg, and Cd: 4.00 × 10^−2^, 3.00 × 10^−1^, 1.4 × 10^−1^, 7.00 × 10^−1^, 3.00 × 10^−4^ and 1.00 × 10^−4^, and 3.00 × 10^−4^ mg/kg/day, respectively; HQ is the Hazard Quotient; AW is the average adult weight (70 kg); and HQ_EFA_ is the Hazard Quotient for fish consumption when a human person aims to obtain from the fish the recommended dose of EFA (risk benefit ratio for fish consumption both metal and EFA, respectively).

The benefit–risk quotient (BRQ) was calculated according to the following equation [44]:BRQ = Q_FA_/Q_T_ Q_FA_ = R_FA/_C_FA_    Q_T_ = (RfD × bw)/c(12)
where R_FA_ (mg/day) is the recommended daily intake of EPA + DHA (250 mg per day) for a healthy adult; C_FA_ (mg/g) is the concentration of EPA + DHA in fish muscle tissue; Q_T_ is the maximum allowable fish consumption related to toxic effects; RfD (mg/kg bw/day) is the Reference Dose of a pollutant; bw is the body weight (70 kg); and c (mg/g) is the content of metals in fish muscle. 

HQ_EFA_ and BRQ values below 1 suggest that the recommended consumption of EPA + DHA does not present a clear risk to human health from the consumption of selected metals through the consumption of fish.

The fatty acids were quantified in g/100 g of edible fish muscles using the following formulae [45,46]:FA (g/100 g) = [(P × FC)/100] × C(13)
where FA is the fatty acid (g/100 g muscles of fish); P is the fatty acid (% of total lipid); FC is the fat content (g/100 g fish muscles); and C is the conversion factor (0.900 for fatty fish).

#### 2.8.7. Hypocholesterolemic/Hypercholesterolemic Ratio (HH) 

The HH was calculated using the following pattern [47,48,49,50,51]:HH = (C18: 1n − 9 + 18: 2(n − 6) + 20: 4(n − 6) + C18: 3(n – 6) + 20: 5(n − 3) + 22: 5(n − 3) + 22: 6(n − 3))/C14: 0 + C16: 0(14)

#### 2.8.8. Nutritive Value Index (NVI) 

The NVI was defined according to the following equation [47]:NVI = C18: 0 + C18: 1/C16: 0(15)

#### 2.8.9. Health-Promoting Index (HPI)

The HPI can be define according to the formulae [28,31]:HPI = ΣUFA/[C12:0 + (4 × C14:0) + C16:0](16)

#### 2.8.10. The de Minimis EPA+DHA:Hg 

The *de minimis* was calculated using the following pattern by Sulimanec Grgec et al. [52].
[RDI(EPA + DHA)/CR)]: [TDI_THg_/CR)×BW](17)
where RDI for EPA + DHA is 250 mg per day; CR is the consumption rate of fish (34.52 g fish per day); TDI is the Tolerable Daily Intake (μg/kg body weight/day); and BW is the average body weight (70 kg).

### 2.9. Statistical Analysis

Data were analyzed using Microsoft Excel and Statistica v. 13.3 (TIBCO Software Inc., Palo Alto, CA, USA). The Pearson correlation coefficient test was used to check for significant relationships between metals and size (the length and weight of the fish), between pairs of metal, and between selected fatty acids and total mercury. The results were considered statistically significant when *p* < 0.05. Principal component analysis (PCA) and figures were made using the Statistica 13.3. (StatSoft, Inc., Tulsa, OK, USA). PCA was performed on the content of elements or fatty acids in muscle tissue depending on the fish.

## 3. Results

### 3.1. Metal Bioaccumulation

The average concentration of copper, zinc, manganese, iron, mercury, and cadmium in the muscles of eels is shown Figure 1b,c. The results are expressed in mg/kg wet weight. The content of these metals in all samples analyzed was 0.371, 23.115, 0.067, 2.011, 0.415, and 0.0012 mg/kg, respectively. The content of magnesium, sodium, calcium and potassium in the muscles of eels is shown in the Figure 1a; it was 11.41, 133.9, 27.57, and 204.7 mg/100 g, respectively. The results are expressed in mg/100 g wet weight. The content of metals in the muscles of fish examined gave rise to the following sequence: K > Na > Ca > Mg > Zn > Fe > Hg > Cu > Mn > Cd (Figure 2).

In most cases, the correlation between the concentration of these metals studied in the muscles of eels and the size of the eels (body weight and total length) was not statistically significant (*p* > 0.05) (Table 1). Positive correlation coefficients (*p* < 0.05) were observed between Hg level in the muscle tissue of eels and body weight (r = 0.4881, *p* = 0.034) and total length (r = 0.4855, *p* = 0.035). Principal component analysis (PCA) plot (2D) showed relationships between variables such as elements, total body length (LT), and body weight (BW) in eels. (Figure 3) Positively correlated variables are grouped together. Negatively correlated variables are placed on opposite sides of the graph (opposite quadrants). The further from the center of the circle the points are, the higher the correlation of the variable with the main component.

Significant positive correlation coefficients were found between metal pairs in muscles of the fish examined (Fe-Mn, Fe-Cu and Zn-Cu), whereas a significant negative correlation was observed between a pair of Mn-Cd (*p* < 0.05) (Table 1).

### 3.2. Human Health Risk Assessment

The THQ, HI, EDI, EWI, and MPI are shown in Table 2. The target hazard quotient (THQ) for Cu, Zn, Mn, Fe, Hg, and Cd in the muscles of eels is presented in Table 2. The fish consumption was 12.6 kg per capita/year (for an adult with a body weight of 70 kg) (Statistical Yearbook of Agriculture 2021 [53]). THQ values below 1 show that there is no non-carcinogenic health risk for the consumer by consuming 34.52 g of fish portion per day. The estimated daily intake of metals (EDI) was 0.1828 μg/body weight (Cu), 11.3989 (Zn), 0.0331 (Mn), 0.9918 (Fe), 0.2045 (Hg), and 0.0006 (Cd) μg/body weight. HI calculated by the sum of the hazard quotients of all metals did not exceed 1; it was 0.7319, which also indicates that the consumption of the tested fish is safe for the consumer from the nutritional point of view. The Metal Pollution Index (MPI) was lower, being 0.2903. The concentration of metal is expressed in microgram per gram of wet weight.

### 3.3. Fat and Fatty Acids Composition

A total of 25 fatty acids, including 7 SFA, 5 MUFA, 7 n-3 PUFA and 6 n-6 PUFA were identified. In the present study, the content of fat (total lipid) in muscles of the European eel ranged between 13.72% and 30.41%, whereas the average value was 21.26% (Table 3). Among SFA, the predominant fatty acid was palmitic C16:0 (19.95%), whereas the most abundant MUFAs was oleic acid C18:1 (37.23%), followed by palmitoleic acid C16:1 (9.17%). Arachidonic C20:4(n-6)AA (2.78%), docosapentaenoic C22:5(n-3)DPA (2.57%), eicosapentaenoic C20:5(n-3)EPA (2.38%), and docosahexaenoic C22:6(n-3)DHA (4.84%) acids were the major components among polyunsaturated fatty acids (Table 3). The muscle tissue of eels contained 30.12% of SFAs, 47.77% of MUFAs, and 22.11% of PUFAs, including 14.16% of n-3 PUFAs and 7.94% of n-6 PUFAs (Table 4). The groups of fatty acids in the muscles of these fish gave rise to the following sequence: MUFAs > SFAs > n-3 PUFAs > n-6 PUFAs (Figure 4). Principal component analysis (PCA) plot (2D) of the relationships between all variables (fatty acid (% of the total fatty acids), total body length (LT), and body weight (BW)) in eels is presented Figure 5. Positively correlated variables are grouped together (PCA plot (2D)) (Figure 5). Negatively correlated variables are placed on opposite sides of the graph (opposite quadrants). The further from the center of the circle the points are, the higher the correlation of the variable with the main component. Hypocholesterolaemic (DFA) and hypercholesterolaemic fatty acids (OFA) in the muscles of fish were as follows: 74.36 and 24.69 (Table 4). Atherogenic index (AI) and thrombogenicity index (TI) were calculated as follows: 0.55 (AI) and 0.41 (TI), whereas flesh-lipid quality index (FLQ) was 7.23. A n-3/n-6 ratio amounting to 1.83 was found in the muscles of European eels. The ratio of the benefits of consuming the recommended intake of DHA + EPA (12.68 mg/g edible portion) (Table 5) to the intake of metals consumption shows that HQE_FA_ or BRQ < one indicates no obvious risk associated with the consumption of fish examined (Table 2), and HQ_EFA_ or BRQ > one suggests such a risk.

The hypocholesterolemic/hypercholesterolemic ratio (HH) was 2.26, while the PUFA/SFA ratio was 0.74. The following NVI and HPI values (2.10 and 1.82, respectively) were found in the muscles of the examined fish, where NVI is the nutritive value index and HPI is health-promoting index (Table 4).

Fatty acids content (expressed as mg/g fish muscles) and the fatty acids ratio (mg)/mercury (µg) in muscle tissue of the European eel are shown in Table 5. Based on the results, the sum of EPA and DHA was found to be 12.68, whereas Σ SFA, Σ MUFA, Σ n-6 PUFA, Σ n-3 PUFA, and Σ PUFA were found to be 51.88, 82.13, 13.48, 24.71, and 38.19, respectively. The studies also determined the ratios of EPA/Hg, DHA/Hg, EPA + DHA/Hg, PUFA/Hg, n-3 PUFA/Hg, and n-6 PUFA/Hg, which were 12.40, 23.96, 36.36, 116.21, 72.66, and 43.56.

The *de minimis* EPA + DHA:Hg for the examined European eels was 17.46. The ratio DHA + EPA/Hg exceeded the established *de minimis* ratio for EPA + DHA:Hg (Table 5). The concentration of mercury and EPA + DHA in muscle tissue of the fish examined were not significantly correlated (*p* = 0.247) (Table 6). Similarly, positive relationships between Hg and the groups EPA, DHA, Ʃ n-3 PUFA, and Ʃ PUFA, were found, but the correlations were not statistically significant (*p* > 0.05). There was a negative correlation between the content of Hg and Ʃ n-6 PUFA; however, it was not statistically significant (*p* > 0.05). The n-3/n-6 ratio increased with the concentration of Hg in the muscles of the European eel (*p* = 0.002). 

## 4. Discussion

The European eel, as a predatory fish, is at the top of the aquatic environment food chain [2]. The factors influencing the toxicity and harmfulness of heavy metals include species, age, sex, size, dietary habit, and preferred habitat of fish, as well as the physical or chemical properties of water, heavy metal interactions with each other, and bioavailability [25]. In addition to the specific toxicity of the pollutants concerned, and to the exposure time and concentration, the effects are influenced by various environmental factors such as pH, oxygen concentration, temperature, and salinity [54]. An example of the fact that the species affects the content of Zn, Mn, Cu, Fe, Hg, and Cd was the research conducted by Rakocevic et al. [55], who marked these metals in the muscles of rudd (*Scardinus knezevici* L.), bleak *(Alburnus scoranza* L.), carp (*Cyprinus carpio* L.), roach (*Rutilus prespensis*), the European eel, and perch (*Perca fluviatilis* L.). The fish were collected from Skadar Lake (Montenegro); however, the differences between the species were not always statistically significant. A highly significant difference in Fe level was observed between eel (3.23 mg/kg) and all the other fish species and followed the following pattern: roach > bleak > carp > rudd > perch > eel. The European eel (0.199 mg/kg) and pikeperch (*Zander lucioperca* L.) contained more Hg, followed by pike (*Esox lucius* L.), bream (*Abramis brama* L.), roach (*Rutilus prespensis*), perch, common carp, and catfish (*Silurus glanis* L.) (France) [56]. The same authors found that the European eel (0.011 mg/kg) had more Cd, followed by roach, bream, common carp, catfish, perch, pike, and catfish. The values of Fe, Hg, and Cd [55] found by the above authors are not in accordance with those found in the eels covered by the research of this paper (Figure 1b, c). There were higher concentrations of Cd (0.006–0.067 mg/kg) and Hg (0.155–0.533 mg/kg) in muscles of the European eel than brown trout (*Salmo trutta* L.) (found in wild ecosystems in Spain), which confirms inter-species differences in metal content [57]. According to the same authors, the greater longevity of the European eel, as well as its different eating and behavioral habits, can explain the differences in the content of heavy metals between this species and brown trout. Polak-Juszczak [58] found that the bioaccumulation of mercury forms depended largely on the feeding habits of species from the southern Baltic Sea, i.e., cod (*Gadus morhua* L.), eel, herring (*Clupea harengus* L.), and sprat (*Sprattus sprattus* L.). In addition, the transfer and biomagnification of mercury in fish influenced food and trophic position. Achouri et al. [59] noted that macronutrients in the muscles of wild and breeding eel from the fish market of Sfax (Tunisia) (Na, K, Mg, and Ca) varied between 12.86 (wild eel) and 225.23 mg/100 g (wild eel), whereas micronutrients such as Fe, Zn, Cu, and Mn ranged from 1.52 (wild eel) to 83.7 mg/kg (breeding eel). According to those authors, the content of elements in the muscle tissue of eel decreased in the order K > Na > Ca > Mg > Zn > Fe > Mn > Cu (wild eel) and K > Na > Mg > Ca > Zn > Fe > Mn > Cu (breeding eel). Morhit et al. [60] found that the order of metal bioaccumulation in eels (Morocco) was Fe > Zn > Cu > Cd. They concluded that this order could be attributed to the different processes of absorption, metabolism and detoxification in the fish. These results are not consistent with the present study (Figure 2). Rakocevic et al. [55] found that the average contents of trace elements in the muscles of fish from Skadar Lake gave the following ranking: Zn > Fe > Mn > Cu > Hg > Cr > As > Ni > Pb > Cd. Based on these data, a decreasing sequence of metals was not observed to be consistent with the results of this work (Figure 2). The research conducted by the Polak-Juszczak and Robak [2] showed that the muscles of eels from various regions of northern Poland contained different amounts of macro- and microelements. It should also be added that among the macroelements, the content of potassium was several times higher than the values of other elements belonging to this group, while among the microelements, zinc was dominant. Similar observations were observed in the current study (Figure 1a,b). Yorulmaz et al. [61], studying the muscles of eels inhabiting the Köyceğiz-Dalyan Lagoon System in the south-west of Turkey, found the following sequence: Fe > Zn > Cu > Mn > Cd > Hg. This sequence does not confirm the current study nor the work of other authors. According to the authors, the content of heavy metals in the muscles of the examined eels depended on the sampling point. The presence of Cd and Pb in eels consumed by the local population and of commercial importance may have resulted from intensive tourism activities (e.g., boat traffic, motor oil, ballast water, and phosphate fertilizers used in agriculture). Rudovica and Bartkevics [62] found no differences in the concentration of Cd, Pb, and Hg in eel muscles among geographic locations; therefore, they did not correlate with agricultural activity, although some of these lakes were located closer to areas actively used for agriculture. Yorulmaz et al. [61] also noted that agriculture occupies an equally important place for the economic value of the study area, such as tourism. Conservative fungicides, mainly used in citrus plantations and greenhouses around the study area, contain Cu, which may have contributed to the Cu contamination of this aquatic environment. Similarly, fertilizers containing Zn and Mn are widely used in agriculture in the study area. The fact that the place of harvesting affects the content of Cu and Cd, but not Zn, in the muscle tissue of eels (Southern France) was also suggested by Amilhat et al. [23]. The contents of Cu, Zn, Mn, Fe, Hg, and Cd in the muscles of the European eels examined differed from those in the Köyceğiz-Dalyan Lagoon System (Cu: 2.435–43.04, Zn: 16.584–70.047, Mn: 0.509–4.841, Fe: BDL-75.13, Hg: 0.066–0.221, and Cd: 0.081–0.425 mg/kg) [61], Latvian lakes (Cu: 0.77–0.92, Zn: 28.00–42.00, Mn: 0.20–0.29, Fe: 8.40–13.20, Hg: 0.13–0.36, and Cd: 0.0051–0.011 mg/kg) [62], Köyceğiz Lagoon System (Cu: 6.32, Zn: 32.92, Hg: 0.14, and Cd: 0.22 mg/kg) [63], Sebou estuary in Morocco (Cu: 0.72, Zn: 46.30, and Fe: 14.80 mg/kg) [64], Belgian River (Yser River) (Cu: 0.0005, Zn: 0.0239, Hg: 0.150, and Cd: 0.0025 mg/kg) [65], Lake Edku (Cu: 4.31 and Zn: 17.84 mg/kg) [66], Spanish rivers (Fererias River) (Cu: 0.210, Hg: 0.533, and Cd: 0.013 mg/kg) [57], Gironde estuary (France) (Cu: 8.7, Zn: 246.7, Hg: 1.19, and Cd: 52.3 ng/g dry weight) [67], Nature Park Hutovo Blato (Bosnia and Herzegovina) (Hg: 0.159 and Cd: 0.02 mg/kg) [68], the lagoons of Fogliano and Caprolace (Italy) (Cu: 0.29, Zn: 30.37, Hg: 0.31, Cd: 0.00, Cu: 0.31, Zn: 28.92, Hg: 0.30, and Cd: 0.00) [69], station Port from Loukkos estuary (Morocco, eastern Atlantic) (Cu: 0.45, Zn: 39.60, Fe: 80.35, and 0.45 dry weight) [60].

It is known that fish are a suitable indicator of heavy metal contamination in the aquatic ecosystem as they occupy different trophic levels and are of different sizes and ages [70]. According to those authors, the concentrations of all analyzed trace metals (including Cu, Fe, Mn, and Zn) in muscle tissues of eels from the Tersakan stream (Mugla, Turkey) increased with eel age, body weight, and total length (*p* < 0.05), i.e., concentrations of trace metals were generally higher in older and larger tissues compared to younger and smaller fish. Similarly, the content of Cd in muscle tissue of eels from the Tersakan stream (Mugla) was positively correlated with age, total length, and weight (r = 0.19, 0.14, and 0.23, *p* < 0.05, respectively) [71]. The results are not in accordance with the observations in fish examined (Table 1). Polak-Juszczak and Robak [2] reported that the concentrations of macro- and microelements (r = from −0.2 to 0.5), with the exception of *p* and Zn, were negatively correlated with eel length and weight. In the muscles of the same species of fish, the subject of this study, the content of Mg, Na, K, Fe, and Mn was positively correlated with body weight (r = from 0.0380 to 0.2555) and total length (r = from 0.0049 to 0.3699) (Table 1), whereas the concentration of Ca, Zn and Cu decreased as body weight (r = from −0.0221 to −0.4030) and total length (from −0.1348 to −0.3305) increased. It should be noted, however, that these correlations were not statistically significant (*p* > 0.05). The content of Hg in muscles of eels (Poland) was positively correlated with body weight (r = 0.61 and total length r = 0.40), whereas there was a negative correlation between the content of Cd in the muscles of eels and total length (r = −0.51) or weight (r = −0.57) [2]. Guhl et al. [72] showed a weak positive correlation with length (*p* = 0.049) and weight (*p* = 0.055) of eels from the North Rhine-Westphalian rivers. In the case of European eels from different French fishing areas, negative correlations with body weight (r = −0.35, *p* < 0.01) and total length (r = −0.37, *p* < 0.01) were found for Cd, but there was no significant correlation for Hg [56]. For European eels in Flemish (Belgian) waterbodies, the effect of length for Hg in muscles was also not statistically significant (*p* = 0.06) [73]. In contrast, the content of Hg in muscles of the eels examined (Table 1) increased as body weight and total length (*p* < 0.05) increased; the correlation between these parameters and Cd was negative but not statistically significant (*p* > 0.05). Significant positive correlations were found for Zn–Fe, Zn–Mn, and Zn–Cd, while significant negative correlations were observed for Zn–Cu in muscles of fish collected from Skadar Lake (Montenegro), including the European eel [55]. Significant positive correlation between Zn–Fe in muscles of eels (Morocco) was reported by Morhit et al. [60]. Genc and Yilmaz [63], studying the muscles of eels in the Köyceğiz Lagoon System, observed a significant positive correlation between the metal pairs Hg–Cd and Cu–Zn (r = 0.461 and 0.420, *p* < 0.001, respectively). However, they did not find significant positive or negative correlations between the following pairs: Hg–Cd (r = 0.096), Cd–Cu (r = 0.084), Hg–Zn (r = −0.076), and Cd–Zn (r = −0.045) (*p* > 0.05), respectively. The observations presented by those authors are in agreement with those of the present study, but only for Cu–Zn. Yorulmaz et al. [61] found a significant positive correlation between Cd and Hg (r = 0.14, *p* < 0.05). The results obtained by these authors are not consistent with the results of the current study (Table 1). 

### Health Risk

Mercury and cadmium values (Figure 3) in the muscles of the European eels did not exceed the maximum level set by the EU Commission Regulation (1.0 and 0.1 mg/kg) [74]. According to Nauen [75], the Zn (Figure 2) and Cu (Figure 3) values in the European eels examined were below the maximum levels (30–150 and 30 mg/kg, respectively). The level of Mn (Figure 3) in the muscles of the fish examined was low and did not exceed the maximum limit of 2.5 mg/kg set by FAO/WHO. The value of Fe in the fish examined (Figure 2) also did not exceed the limit of 50 set by Zencir [76]. The THQ for the species of fish covered by this study was below one. These data may indicate that the consumption of these fish brings health benefits and consumers are safe. 

All eels from North Rhine-Westphalia had mercury levels below the EU limit (1.0 mg/kg bw) but above the EQS of 20 μg/kg, a measure of secondary poisoning [72]. Similarly, the concentrations of Cd and Hg in the muscles of eels from the Mar Menor lagoon (SE Spain) are below the maximum levels permitted by European legislation [77]. Marwa and Mazrouh [66] found that heavy metal contamination is one of the significant factors that reduces water quality. This consequently affects the diversity of fish and their consumption from a contaminated site poses a greater risk to human health. The same authors noted that the bioaccumulation of metals in the organs of eels from Lake Edku in the north of the Nile Delta (i.e., Cu, Zn, Pb, and Ni) did not exceed the acceptable limits set by the FAO. Similarly, Rakocevic et al. [55], determining trace elements in the muscles of six species of fish, including eel, from Lake Skadar (Montenegro), found that Hg, Pb, Cd, and As were below the maximum permissible levels of many recognized international standards, indicating that all six species can be recommended for the human diet. Due to the low accumulation of heavy metals, the Latvian freshwater ecosystem is relatively clean and the lakes under study are suitable for stocking with eel populations [62]. Although the Cd and Pb content in the muscles of the European eel was below the maximum allowable threshold set by European Union legislation, 7% of the fish analyzed showed values above the mercury threshold values. By contrast, the mercury content exceeded the threshold value for all analyzed eels included in the Water Framework Directive Environmental Quality Standards. Linde et al. [57] observed that seven eels caught in two wild Spanish rivers, which accounted for 12.1%, had muscle mercury levels exceeding the limits established by Spanish regulations for fish intended for human consumption (Real Decreto 157/1977 and 90/1973). Due to the fact that these fish have been caught in places where fish populations are exploited by sport fishing (usually angling) and catches are eaten fresh, more care should be taken when consuming wild eel, as it poses the risk of people consuming heavy metals. The concentrations of trace metals in the muscles of eels did not exceed the WHO/FAO, EPA, IAEA-407, and TFC guidelines. Therefore, consumption of these fish may be safe for human health despite possible heavy metal contamination [70].

According to Polak-Juszczak and Robak [2], consumers appreciate the organoleptic properties of eel meat, especially in the countries where this fish is found, because it is highly valued by many consumers for the taste and texture of its meat. The lipid profiles of the eels indicate their high nutritional quality [78]. The percentage of total fat in the wild eel (22.29%) was higher than that of the breeding one (16.11%) [59]. The fat content in the European eel (Spain) (4.95–10.22%) was the most variable parameter and was directly related to the weight of the fish [78]. The value of total lipids in the muscles of the European eels examined (Table 3) was higher than those observed by the above authors [78]. Among SFA, the major fatty acids in the European eel from the Croatian coast of the Adriatic Sea, including the Neretva River estuary, were palmitic (C16:0), myristic (C14:0), and stearic acids (C18:0). The most abundant MUFA was oleic acid (C18:1) and the second most abundant fatty acid was palmitoleic acid (C16:1), whereas among PUFA, the predominant fatty acid was arachidonic (C20:4 n-6), followed by EPA and DHA [79]. According to Can Tuncelli et al. [80], the major fatty acids in muscles European eels (Orontes River, Turkey) at all seasons were C18:0, C16:0, C16:1, C18:1 (n-9), C22:6 n-3 (DHA), and C18:2 (n-6). The findings presented by these authors showed that European eels from the Orontes River have high nutritional benefits with their high-quality lipid and fatty acid content [80]. The major SFA, MUFA, and PUFA in eels from the fish market (Tunisia) were C16:0, C18:0, C16:1, C18:1, C20:4 n-6, EPA, and DHA. The farmed and wild fillets of this species showed important n-3/n-6 ratios (3.28 and 1.3) and the high stability of their oils makes them suitable for the prevention of ischemic heart disease [59]. The most abundant fatty acids in all fish, including eel, from lake Trasimeno (Italy) was 16:0, 14:0, 18:0, 18:1n-9, 16:1n-7, 18:2n-6, 20:4n-6, EPA, and DHA [44]. These findings are in agreement with those of present study (Table 3). These authors also found that the content of all fatty acids varied significantly among species, whereas the n-3/n-6 ratio ranged between 0.96 (eel) and 3.52 (pike). BRQ was below one, which allows us to say that the species studied by Branciari et al. [44] can be regularly consumed by people without posing a significant threat to their health with regards the presence of pollutants (including Hg and Cd). Therefore, consumption of fish from Lake Trasimeno should be encouraged, all the more so because they allow consumers to meet the recommended levels of EPA and DHA. Can Tuncelli et al. [80] observed that the composition of some fatty acids of European eels harvested from the Orontes River (Hatay, Turkey) varied significantly (*p* < 0.05) between seasons, and the n-3/n-6 ratio ranged from 1.14 to 1.72. The authors demonstrated a consistent, high-quality lipid content in terms of nutritional value, which is important for product quality in this fish, which offers tasty and attractive products with a high content of high-quality fats. The n-3/n-6 ratio in eels in local fish markets (Pakistan) was less than one, whereas the sum of EPA and DHA was 1.755 [81]. These results are not accordance with those present in the fish examined (Table 5). Prigge et al. [82] determined fatty acids in eels from a commercial fish farm (Germany) that were fed on freshwater and marine diets. On this basis, he confirmed the distinction between eels feeding in marine and freshwater habitats. According to these authors [82], the n-3/n-6 ratio in the muscles of eels reared on diets of roach was 2.19, whereas that for eels reared on diets of herring was 3.20. Carnivorous fish had higher contents of EPA + DHA (19.9 and 24.2%) than herbivorous fish (14.9 and 12.9%) from both aquatic ecosystems (freshwater and marine), whereas n-3/n-6 ratios were higher in herbivorous freshwater fish (3.1) and carnivorous marine fish (8.2) [83]. Habitat was a major driver of differences in the fatty acids composition of muscle tissue in Norwegian eels [84]. According to these authors, EPA + DHA was 19.7 (seawater), 23.9 (brackish water), and 14.4 (freshwater), whereas the ratio of n-3/n-6 was 5.8, 7.5 and 1.8, respectively. The n-3/n-6 ratio in the fish examined (Table 5) was similar to that observed in the muscles of Norwegian freshwater eels. Parzanini et al. [85] found that the lipid and fatty acid information provided a better understanding of the use of different habitats and the overall ecology of this critically endangered species.

Zula and Desta [48] found that muscles of raw Nile tilapia (*Oreochromis niloticus* Linnaeus, 1758) from Hawassa Lake (Ethiopia) had a high NVI (1.23) and HH ratio (1.71), whereas the fried fish from the Hawassa fried fish selling market had a high AI (1.01) and TI (1.58). These values are not consistent with those in the muscles of the European eel examined (Table 4). The HH, AI, and TT indices in other species from Croatian sea farms (sea bream (*Sparus aurata* L.), sea bass (*Dicentrarchus labrax* L.), dentex (*Dentex dentex* L.), and turbot (*Scophthalmus maximus* L.)) were as follows: 0.91–3.94, 0.32–1.10, and 0.35–0.97, respectively [86]. Based on this, the authors concluded that all fish species, except turbot, met the recommended AI, TI, and ratio of HH. The PUFA/SFA ratio in the same species was between 0.30 and 1.15. 

The PUFA/SFA in six of the most frequently fished species inhabiting the Vistula Lagoon (Poland) exceeded the recommended minimum value (0.63–2.04) [87]. The same author noted that the UK Department of Health recommends a minimum value of this ratio of 0.45. Chen et al. [28] reported that the PUFA/SFA ratio is usually used to assess the impact of diet on cardiovascular health (CVH). The PUFA/SFA ratio of European eel muscle tissue ranged between 0.48 and 0.52, and it did not differ significantly in relation to weight of the fish [78]. The same authors found that HH, AI, and TI were 1.80–2.05, 0.64–0.66, and 0.58–0.63, respectively. In contrast to HH, the values of the AI and TI were higher than those in the present study (Table 4). However, low values of the AI and TI (<one) are assumed to be beneficial to human health, and the consumption of eels may reduce the risk of coronary heart disease. Comparing the PUFA/SFA ratio to the HH ratio, it was noted that HH may more accurately reflect fatty acid composition on CVD [28]. According to Achouri et al. [59], the nutritional quality index (n-3/n-6, DHA/EPA, AI, TI, and HH), indicated that both farmed and wild eels could be eaten due to the good quality of the lipid fraction. TI is defined as the relationship between procoagulant (saturated) and anticoagulant fatty acids (MUFA, PUFA-n6, and PUFA-n3) [29,30]. According to the AI equation, all unsaturated fatty acids are considered equally effective in reducing the risk of atherosclerosis. The authors called the inverse of AI the health-promoting index (HPI) [30,31]. The Health-Promoting Index (HPI) was proposed by Chen et al. [31]. The authors assessed the nutritional value of dietary fats, focusing on the effect of fatty acid composition on CVD [28]. Karsli [88] observed that the HPI in fish oil supplements (Turkey) varied between 1.30 and 5.47. Bazarsadueva et al. [89] examined the muscle tissue of bream from Lake Kotokel (Western Transbaikalia) collected in 2019 and calculated lipid quality indices such as AI, TI, FLQ, OFA, DFA, and HH at the level 0.44, 0.38, 14.4, 25.0, 72.2, and 2.9. The values observed by these authors were similar to those found in the muscles of the European eel examined in this study (Table 4). García-Gallego et al. [90] reported that the higher the FLQ value, the better the nutritional standard of the lipid source, and in eels (Spain) they reported the following range of values for this indicator: 12.7–24.5.

On the one hand, mercury in fish is a potential threat to human health. On the other hand, fish is a rich source of EPA and DHA, which are known to be beneficial for the function of humans. However, the risk assessment may be confounded by selenium, which is a mercury antagonist [91]. Therefore, selenium should also be included in fish in future studies. However, among freshwater fish species, including Burbot (*Lota lota*; also known as Loche or Mariah), Cisco (*Coregonus artedi*; also known as Herring), Lake Trout (*Salvelinus namaycush*), Lake Whitefish (*Coregonus clupeaformis*), Longnose Sucker (*Catostomus catostomus*), Northern Pike (*Esox lucius*; known locally as Jackfish), Walleye (*Sander vitreus*; also known as Pickerel), and White Sucker (*Catastomus commersoni*), which were harvested from eight lakes in the Dehcho Region of the Northwest Territories, only Longnose Sucker exceeded the *de minimis* ratio for EPA + DHA:THg [92]. According to Sulimanes Grgec et al. [52], the analyzed fish from the Adriatic Sea (Croatia), except tuna (*Thunnus thynnus* L.) (raw fillets, canned meat) and wild-caught Gilthead seabream (*Sparus aurata* L.), exceeded the *de minimis* ratio of 20.7:1. Those authors also observed the highest EPA + DHA:THg ratio in pilchard (*Sardina pilchardus* Walb.), as wild fish from Adriatic Sea (140:1). The same observation was found in the European eel examined (Table 5). Strong negative correlations were found between Hg and total n-3 PUFA in the muscle tissue of burbot (*Lota lota* L.) (r = −0.670, *p* < 0.0001), Northern pike (r = −0.479, *p* < 0.0001), and walleye (*Sander vitreus* Mitchill) (r = −0.446, *p* < 0.0001) [92], which is not in agreement with the results presented in this paper (Table 6). The same authors observed strong negative relationships between Hg and EPA + DHA in the muscles of walleye harvested from Kakisa Lake, but not between Hg and Ʃ n-3 PUFA or Hg and Ʃ PUFA. Strandberg et al. [91] also found that the content of Hg for European perch from ten lakes in Eastern Finland (r = −0.630, *p* < 0.001) decreased as EPA + DHA increased. These results are not consistent with those observed in the present study (Table 6). 

## 5. Conclusions

The muscles of the examined eels are a good source of minerals, which should encourage the consumption of this fish, despite the fact that it belongs to the species of oily fish. In the literature, it is stated that fish, especially oily ones and those high up the food chain (for example, eel) can be contaminated with mercury. The level of Hg in the muscles of the European eel examined only depended on the size of these fish, but did not exceed the permissible standards set by the Commission Regulation (EC) No 629/2008. Similarly, the content of other heavy metals did not exceed the maximum limits. In addition, the THQ index showing a non-carcinogenic risk to consumer health was below one, indicating that the consumption of meat of a given species is safe from a health point of view. The meat of the examined fish is also a rich source of fatty acids with a positive dietary effect for human health, which is indicated by low OFA, AI, and TI indices. and a high DFA index. Based on the HQ_EFA_ and BRQ indices, which were less than one, it can be concluded that the intake of the recommended dose of EPA + DHA (250 mg/day) and the concentration of mercury calculated for a person weighing 70 kg does not pose an obvious threat to human health. However, it should be remembered that the ratio of mercury to EPA + DHA exceeded the established *de minimis* ratio for EPA + DHA:Hg. Therefore, research in this direction should be conducted. Similarly, the content of toxic metals in the muscles of eels, as well as in their environment, should still be monitored, the more so because the eel, as a migratory catadromous fish which travels long distances, is a common predator at the top of the food chain.

## Figures and Tables

**Figure 1 ijerph-20-02257-f001:**
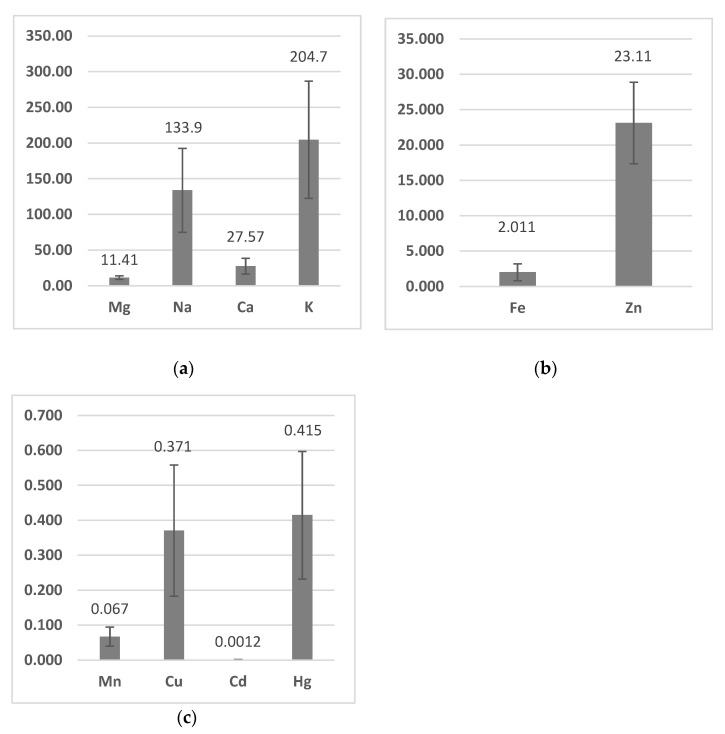
The content of metals (±standard deviation) (**a**) Mg, Na, Ca and K (mg/100 wet weight), (**b**) Fe and Zn (mg/kg wet weight), (**c**) Mn, Cu, Cd and Hg (mg/kg wet weight) in muscles of the European eel (*Anguilla anguilla* L.).

**Figure 2 ijerph-20-02257-f002:**
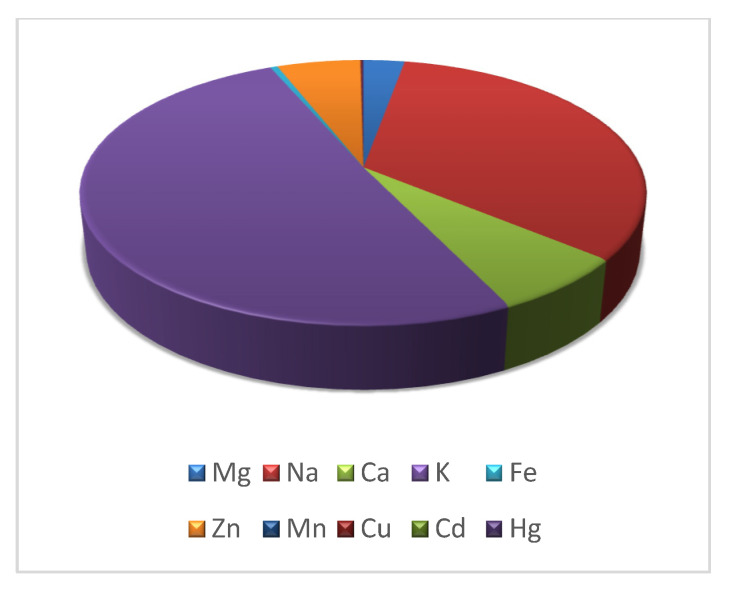
The content of metals in muscles of the European eel (*Anguilla anguilla* L.). Mg, Ca, Na, and K (mg/100 g wet weight); Cu, Zn, Mn, Fe, Hg, and Cd (mg/kg wet weight).

**Figure 3 ijerph-20-02257-f003:**
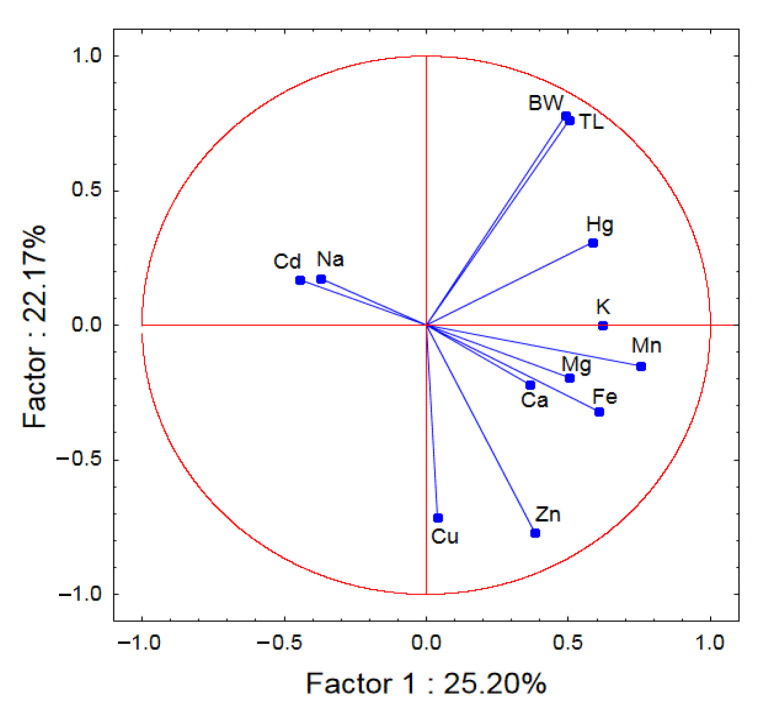
Principal component analysis (PCA) plot (2D) of the relationships between all variables (elements, total body length (LT), and body weight (BW)) in the European eel (*Anguilla anguilla* L.).

**Figure 4 ijerph-20-02257-f004:**
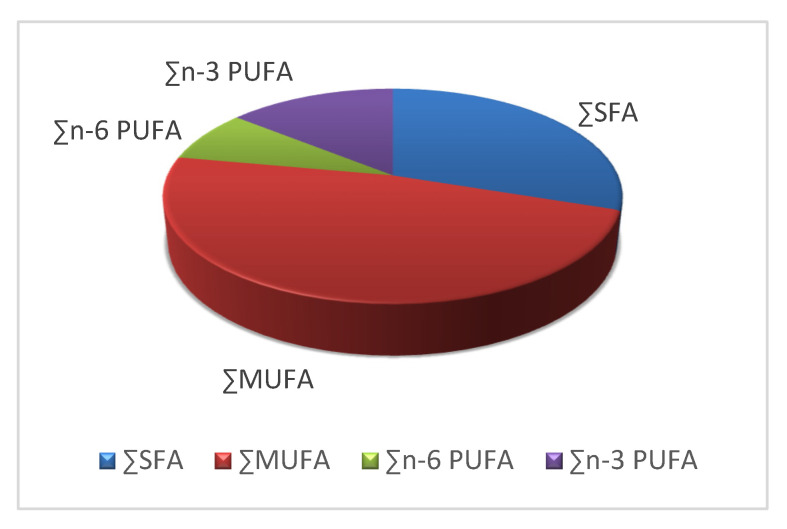
The percentage content of fatty acids (%) in muscles of the European eel (*Anguilla anguilla* L.).

**Figure 5 ijerph-20-02257-f005:**
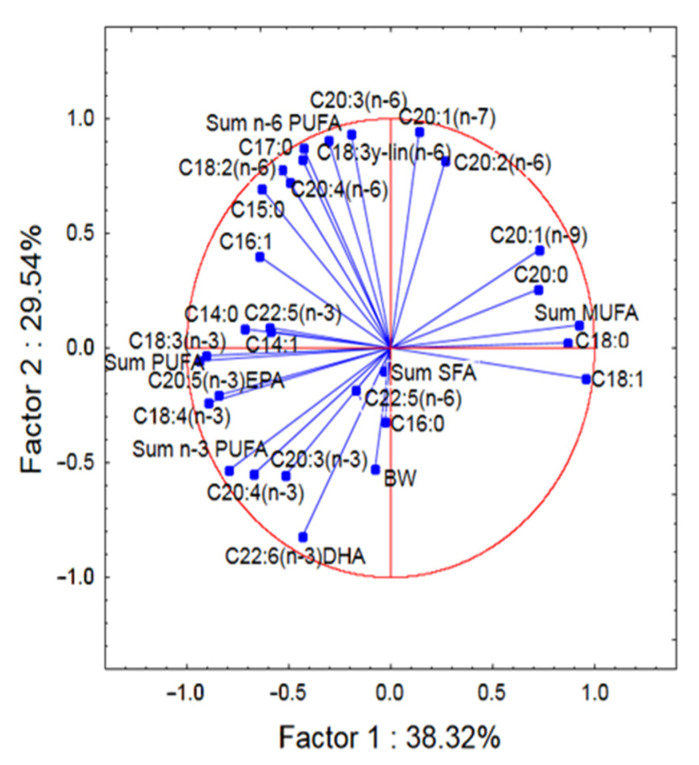
Principal component analysis (PCA) plot (2D) of the relationships between all variables (fatty acid (% of the total fatty acids), total body length (LT), and body weight (BW)) in muscles of the European eel (*Anguilla anguilla* L.).

**Table 1 ijerph-20-02257-t001:** Linear correlation coefficients (r) between metal content in the muscles of eels (*Anguilla anguilla* L.) and body weight or total length.

	Weight	Length	Mg	Na	Ca	K	Fe	Zn	Mn	Cu	Cd	Hg
weight	-	0.9130	0.1847	0.0893	−0.0221	0.1875	0.0380	−0.2892	0.2555	−0.4030	−0.3609	0.4884
		*p* = 0.000	*p* = 0.839	*p* = 0.716	*p* = 0.928	*p* = 0.442	*p* = 0.877	*p* = 0.230	*p* = 0.291	*p* = 0.087	*p* = 0.141	*p* = 0.034
length	0.9130	-	0.0049	0.0424	−0.1348	0.1632	0.1408	−0.3305	0.3699	−0.3112	−0.0542	0.4855
	*p* = 0.000		*p* = 0.984	*p* = 0.863	*p* = 0.582	*p* = 0.504	*p* = 0.565	*p* = 0.167	*p* = 0.119	*p* = 0.195	*p* = 0.826	*p* = 0.035
Mg	0.1847	0.0049	-	−0.0554	0.4019	0.3912	0.2373	0.3480	0.2101	−0.5638	−0.0243	0.2426
	*p* = 0.839	*p* = 0.984		*p* = 0.822	*p* = 0.088	*p* = 0.098	*p* = 0.328	*p* = 0.144	*p* = 0.388	*p* = 819	*p* = 0.921	*p* = 0.317
Na	0.0893	0.0424	−0.0554	-	0.1119	−0.6420	−0.1968	−0.2994	−0.0558	0.1184	0.1764	−0.1456
	*p* = 0.716	*p* = 0.863	*p* = 0.822		*p* = 0.648	*p* = 0.003	*p* = 0.419	*p* = 0.213	*p* = 0.821	*p* = 0.629	*p* = 0.470	*p* = 0.552
Ca	−0.0221	−0.1348	0.4019	0.1119	-	0.1259	0.1766	0.1545	0.4425	0.0860	0.0860	0.0401
	*p* = 0.928	*p* = 0.582	*p* = 0.088	*p* = 0.648		*p* = 0.608	*p* = 0.469	*p* = 0.528	*p* = 0.058	*p* = 0.726	*p* = 0.726	*p* = 0.871
K	0.1875	0.1632	0.3912	−0.6420	0.1259	-	0.1420	0.2418	0.2189	−0.3365	−0.4170	0.1402
	*p* = 0.442	*p* = 0.504	*p* = 0.098	*p* = 0.003	*p* = 0.608		*p* = 0.562	*p* = 0.319	*p* = 0.368	*p* = 0.159	*p* = 0.076	*p* = 0.567
Fe	0.0380	0.1408	0.2373	−0.1968	0.1766	0.1420	-	0.3332	0.5611	0.4809	0.0502	0.4252
	*p* = 0.877	*p* = 0.565	*p* = 0.328	*p* = 0.419	*p* = 0.469	*p* = 0.562		*p* = 0.163	*p* = 0.012	*p* = 0.037	*p* = 0.838	*p* = 0.070
Zn	−0.2892	−0.3305	0.3480	−0.2994	0.1545	0.2418	0.3332	-	0.2674	0.6232	−0.3482	−0.0004
	*p* = 0.2300	*p* = 0.167	*p* = 0.144	*p* = 0.213	*p* = 0.528	*p* = 0.319	*p* = 0.163		*p* = 0.268	*p* = 0.004	*p* = 0.144	*p* = 0.999
Mn	0.2555	0.3699	0.2101	−0.0558	0.4425	0.2189	0.5611	0.2674	-	0.2730	−0.4577	0.2304
	*p* = 0.291	*p* = 0.119	*p* = 0.388	*p* = 0.821	*p* = 0.058	*p* = 0.368	*p* = 0.012	*p* = 0.268		*p* = 0.258	*p* = 0.049	*p* = 0.343
Cu	−0.4030	−0.3112	−0.5638	0.1184	0.0860	−0.3365	0.4809	0.6232	0.2730	-	−0.0348	−0.0096
	*p* = 0.087	*p* = 0.195	*p* = 819	*p* = 0.629	*p* = 0.726	*p* = 0.159	*p* = 0.037	*p* = 0.004	*p* = 0.258		*p* = 0.888	*p* = 0.969
Cd	−0.3609	−0.0542	−0.0243	0.1764	0.0860	−0.4170	0.0502	−0.3482	−0.4577	−0.0348	-	0.01392
	*p* = 0.141	*p* = 0.826	*p* = 0.921	*p* = 0.470	*p* = 0.726	*p* = 0.076	*p* = 0.838	*p* = 0.144	*p* = 0.049	*p* = 0.888		*p* = 0.955
Hg	0.4884	0.4855	0.2426	−0.1456	0.0401	0.1402	0.4252	−0.0004	0.2304	−0.0096	0.01392	-
	*p* = 0.034	*p* = 0.035	*p* = 0.317	*p* = 0.552	*p* = 0.871	*p* = 0.5671	*p* = 0.070	*p* = 0.999	*p* = 0.343	*p* = 0.969	*p* = 0.955	

*p*—significant level.

**Table 2 ijerph-20-02257-t002:** The hazard quotient calculated for metal content in the muscle tissue of fish.

	Cu	Zn	Mn	Fe	Hg	Cd
RfD (mg/kg/day)	4.00 × 10^−2^	3.00 × 10^−1^	1.4 × 10^−1^	7.00 × 10^−1^	3.00 × 10^−4^	1.00 × 10^−4^
EDI	0.1828	11.3989	0.0331	0.9918	0.2045	0.0006
EWI	1.2798	79.7923	0.2317	6.9428	1.4317	0.0041
THQ	0.0046	0.0380	0.0002	0.0014	0.6818	0.0059
HI	0.7319					
MPI	0.2903					
HQ_EFA_	0.0026	0.0217	0.0001	0.0008	0.3897	0.0034
BRQ	0.0026	0.0217	0.0001	0.0008	0.3897	0.0034

EDI—Estimated Daily Intake (μg/kg body weight/day); EWI—Estimated Weekly Intake (μg/kg body weight/week); THQ—Target Hazard Quotient; RfD—Oral Reference Dose (mg/kg/day); HI—Hazardous Index; MPI—Metal Pollution Index, HQ_EFA_; and BRQ—the benefit–risk quotient.

**Table 3 ijerph-20-02257-t003:** Lipid content (%) and fatty acids composition in muscles of the European eel (*Anguilla anguilla* L.).

	Mean	SD	Systematic Name	Trivial Name	Mean	SD
fat	21.26	5.10				
	expressed as % of the total fatty acids				expressed as mg/g fish muscles	
C12:0	0.12	0.04	dodecanoic	lauric	0.21	0.06
C14:0	4.62	0.53	tetradecanoic	myristic	7.96	2.18
C15:0	0.37	0.08	pentadecanoic	pentadecylic	0.63	0.19
C16:0	19.95	1.00	hexadecanoic	palmitic	34.40	8.51
C17:0	0.46	0.07	heptadecanoic	margaric	0.78	0.18
C18:0	4.48	0.50	octadecanoic	stearic	7.70	2.00
C20:0	0.12	0.03	eicosanoic	arachidic	0.21	0.04
C14:1	0.20	0.04	*cis*-9-tetradecenoic	myristoleic	0.34	0.11
C16:1	9.17	1.38	*cis*-9-hexadecenoic	palmitoleic	15.78	4.61
C18:1	37.23	3.33	*cis*-9-octadecenoic	oleic	64.05	16.22
C20:1 (n-7)	0.15	0.07	*cis*-7-eicosenoic	gadoleic	0.24	0.09
C20:1 (n-9)	1.02	0.17	*cis*-9-eicosenoic	gadoleic	1.72	0.31
C18:2(n-6) LA	3.37	0.65	*cis,cis*-9,12-octadecadienoic	linoleic	5.74	1.53
C18:3γ-lin (n-6)	0.16	0.04	*cis*-6,*cis*-9,*cis*-12-octadecatrienoic acid	γ-linolenic	0.27	0.08
C20:2(n-6)	0.55	0.22	*cis*-11,*cis*-14- eicosadienoic	eicosadienoic	0.90	0.28
C20:3(n-6)	0.38	0.12	*cis*-8,*cis*-11,*cis*-14-eicosatrienoic	dihomo-γ-linolenic acid	0.64	0.19
C20:4(n-6) AA	2.78	0.39	*cis*-5,*cis*-8,*cis*-11,*cis*-14-eicosatetraenoic	arachidonic	4.71	1.01
C22:5(n-6)	0.71	0.11	*cis*-4,*cis*-7,*cis*-10,*cis*-13,*cis*-16- docosapentaenoicdocosapentaenoic acid	docosapentaenoic	1.22	0.32
C18:3(n-3) ALA	2.21	0.41	*cis*-9,*cis*-12,*cis*-15-octadecatrienoic	α-linolenic	3.85	1.29
C18:4 (n-3)	0.30	0.08	*cis*-6,*cis*-9,*cis*-12,*cis*-15-octadecatetraenoic acid	stearidonic	0.53	0.22
C20:3(n-3)	0.41	0.07	*cis*-11,*cis*-14,*cis*-17-eicosatrienoic	eicosatrienoic	0.72	0.24
C20:4(n-3)	1.44	0.30	*cis*-8,*cis*-11,*cis*-14,*cis*-17-eicosatetraenoic acid	eicosatetraenoic	2.52	0.91
C20:5(n-3) EPA	2.38	0.51	*cis*-5,*cis*-8,*cis*-11,*cis*-14,*cis*-17-eicosapentaenoic	eicosapentaenoic	4.16	1.49
C22:5(n-3)DPA	2.57	0.37	*cis*-7,*cis*-10,*cis*-13,*cis*-16,*cis*-19-docosapentaenoic	docosapentaenoic	4.40	1.13
C22:6(n-3)DHA	4.84	1.12	*cis*-4,*cis*-7,*cis*-10,*cis*-13,*cis*-16,*cis*-19-docosahexaenoic	docosahexaenoic	8.51	3.25

SD—standard deviation; LA—linoleic acid (C18:2); AA—arachidonic acid (C20:4); ALA—α-linolenic acid (C18:3); EPA—eicosapentaenoic acid (C20:5); DPA—docosapentaenoic C22:5(n-3); and DHA —docosahexaenoic (C22:6).

**Table 4 ijerph-20-02257-t004:** Fatty acids composition (%) and indices of human health in muscles of the European eel (*Anguilla anguilla* L.).

	Mean	SD
EPA + DHA	7.22	1.46
n-3/n-6	1.83	0.39
Σ SFA	30.12	1.21
Σ MUFA	47.77	2.59
Σ n-6 PUFA	7.94	1.25
Σ n-3 PUFA	14.16	2.30
Σ PUFA	22.11	2.51
OFA	24.69	1.25
DFA	74.36	1.27
AI	0.55	0.04
TI	0.41	0.04
FLQ	7.23	1.46
PUFA/SFA	0.74	0.09
HH	2.26	0.18
NVI	2.10	0.24
HPI	1.82	0.15

SD—standard deviation; Ʃ SFA—saturated fatty acid; Ʃ MUFA—monounsaturated fatty acid; Ʃ n-6 PUFA—polyunsaturated fatty acid; Ʃ n-3 PUFA—polyunsaturated fatty acid; EPA—eicosapentaenoic acid (C20:5); DHA—docosahexaenoic (C22:6); AI—index of atherogenicity; TI—index of thrombogenicity; FLQ—flesh-lipid quality; OFA—hypercholesterolemic fatty acids; DFA—hypocholesterolemic fatty acids; HH— hypocholesterolemic/hypercholesterolemic ratio; NVI—nutritive value index; and HPI—health-promoting index.

**Table 5 ijerph-20-02257-t005:** Fatty acids composition (expressed as mg/g fish muscles) and fatty acids ratio (mg)/mercury (µg) in muscles of the European eel (*Anquilla anquilla* L.).

	Mean	SD
EPA + DHA	12.68	4.60
Σ SFA	51.88	12.61
Σ MUFA	82.13	19.65
Σ n-6 PUFA	13.48	2.99
Σ n-3 PUFA	24.71	8.09
Σ PUFA	38.19	10.52
Ratio n-3/n-6	1.83	0.39
Ratio EPA/Hg	12.40	6.74
Ratio DHA/Hg	23.96	11.90
Ratio EPA + DHA/Hg	36.36	18.22
Ratio PUFA/Hg	116.21	67.55
Ratio n-3 PUFA/Hg	72.66	38.78
Ratio n-6 PUFA/Hg	43.56	30.13
*de minimis* EPA + DHA:Hg	17.46	17.05

SD—standard deviation; Ʃ SFA—saturated fatty acid; Ʃ MUFA— monounsaturated fatty acid; Ʃ n-6 PUFA—polyunsaturated fatty acid; Ʃ n-3 PUFA—polyunsaturated fatty acid; EPA—eicosapentaenoic acid (C20:5); DHA—docosahexaenoic (C22:6); the *de minimis* ratio—EPA + DHA:Hg.

**Table 6 ijerph-20-02257-t006:** Linear correlation coefficients (r) between fatty acids (expressed as mg/g fish muscles) in the muscles of eels (*Anguilla anguilla* L.) and total mercury, expressed as µg/g wet weight.

	r	*p*
C20:5(n-3)EPA	0.0644	0.793
C22:6(n-3)DHA	0.3645	0.125
Σ n-6 PUFA	−0.3250	0.175
Σ n-3 PUFA	0.2106	0.387
Σ PUFA	0.0697	0.777
EPA + DHA	0.2789	0.247
Ratio n-3/n-6	0.6584	0.002

Ʃ n-6 PUFA—polyunsaturated fatty acid; Ʃ n-3 PUFA—polyunsaturated fatty acid; EPA—eicosapentaenoic acid (C20:5); DHA—docosahexaenoic (C22:6).

## Data Availability

Not applicable.

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
