# Peer review of "Evaluation of Chemical Elements, Lipid Profiles, Nutritional Indices and Health Risk Assessment of European Eel (Anguilla anguilla L.)"

_ijerph, 2023, doi:10.3390/ijerph20032257_

Round 1
Reviewer 1 Report
The data source and data collection confirm the suitability of the methodology and analytical procedures used in this study. The interpretation of the obtained results is consistent with the current approach in this field.
I suggest some changes on the manuscript and address of the following concerns:
Line 19-21 please rephrase
The introduction is too long and resonant with information covered extensively in the literature I suggest to revise
Could you explain why for the EDI was used 60 kg body weight
Line 431-446 The part thus structured is not suitable for discussions please revise
Author Response
Responses to reviews
Reviewer #1
Line 19-21 please rephrase
The sentence was changed as suggested Reviewer.
The introduction is too long and resonant with information covered extensively in the literature I suggest to revise
The introduction was changed as suggested Reviewer.
Could you explain why for the EDI was used 60 kg body weight
We suggested data from the manuscript: Polak-Juszczak, L.; Nermer, T. Methylmercury and Total Mercury in Eels, Anguilla anguilla, from Lakes in Northeastern Poland: Health Risk Assessment. EcoHealth 2016, 13(3), 582-590. https://doi.org/10.1007/s10393-016-1139-3
In most manuscripts for EDI was used 70 kg body weight.
The sentence was changed in the Section “Material and methods”, “Results”, “Discussion” and Table, as suggested Reviewer.
Line 431-446 The part thus structured is not suitable for discussions please revise
The sentence was changed as suggested Reviewer.

Reviewer 2 Report
Title should be revised to make it more clear and comprehensive
Pie charts should be include in results
disscussion need more refrences
Author Response
Responses to reviews
Reviewer #2
Title should be revised to make it more clear and comprehensive
The Title was changed as suggested Reviewer.
Pie charts should be include in results
Four pie charts have been introduced in the "Results" section.
Figure 2. The content of metals in muscles of European eel (Anguilla anguilla L.).
Mg, Ca, Na and K (mg/100 g wet weight), Cu, Zn, Mn, Fe, Hg and Cd (mg/kg wet weight),
Figure 3. Principal component analysis (PCA) plot (2D) relationships between all variables (elements, total body length (LT), body weight (BW)) in European eel (Anguilla anguilla L.).
Figure 4. The percentage content of fatty acids (%) in muscles of European eel (Anguilla anguilla L.).
Figure 5. Principal component analysis (PCA) plot (2D) relationships between all variables (fatty acid (% of the total fatty acids), total body length (LT), body weight (BW)) in in muscles of European eel (Anguilla anguilla L.).
disscussion need more references
The Section “Discussion” added more references as suggested Reviewer.

Reviewer 3 Report
This research deals with a study that evaluates the chemical elements, fatty acids, and nutritional indices of the Muscles of European eel (Anguilla anguilla L.).
The topic is within the scope of the Journal, the topic is interesting, the manuscript is well-written, and the analyzed parameters are interesting. However, major revisions are requested. Please see the following comments.
Abstract
Background: The justification of the study is lacking, and the purpose of the study is not highlighted. Please consider adding them.
Lines 19-21. This sentence is not finished: "which indicates that a person weighing 60 kg and daily consuming 250 mg of EPA + DHA with mercury content in the muscles of eel (0.415 mg/kg) ….".
Line 22. This sentence, "The fatty acids indices (OFA, DFA, AI and TI) were also studied.", doesn't show any result.
Please delete (2), (3) and (4) from the abstract.
Introduction
Lines 64-66. Please consider rewriting this sentence "The metals belong to the group "trace elements", i.e. required in trace quantities (e.g. Fe, Zn, Cu and Mn) are also classified as heavy metals." for a better understanding.
Lines 66-68. Please consider rewriting this sentence "Besides these elements, heavy metals are also included chromium, cadmium, arsenic, lead, nickel, mercury and selenium." for a better understanding.
Lines 92-93. "According to these authors …". Please indicate which authors you are referring to.
Line 99-100. Please consider rewriting this sentence "Fish, especially fatty fish contain significant also contain the very important n-3 99 polyunsaturated fatty acids." for a better understanding.
Line 103. "According to these authors …". Please indicate which authors you are referring to.
Lines 113-114. Please indicate the references for the 4:1 and 10:1 ratios in the following sentence "The recommended ratio of n-6 to n-3 typically ranges from 4:1 (or less) to 10:1."
Lines 117-118. Please reformulate this sentence "In the literature, the nutritional quality indices are often found." for a better understanding.
Lines 118-121. Please indicate the references for this sentence "This may be related to the fact that all PUFAs in the diet can lower LDL-C, i.e. low-density lipoprotein cholesterol, and lower serum cholesterol, while all SFAs contribute to high serum cholesterol."
Lines 126-128. Please reformulate this sentence "Ulbricht and Southgate [40] and Chen et al. [41], as the inverse of the atherogenic index, named the HPI, which focuses on the effect of fatty acids composition on CVD." for a better understanding.
Lines 128-130. Please reformulate this sentence "Thrombogenicity index (TI) defined as the relationship between the pro-thrombogenic FAs (C12:0, C14:0, and C16:0) and the anti-thrombogenic FAs (MUFAs and the n-3 and n-6 PUFAs) [38,40]." for a better understanding.
Lines 136-137 show the experimental design, not the aim of the study. Also, the originality of the study is not clearly stated.
Materials and Methods
Lines 164-165. "Muscle tissue (without skin) from the dorsal part were prepared from one specimens". Was the muscle sample collected from only one specimen?
Please indicate the conditions of samples transportation to the laboratory and the scarification of fish.
Author Response
Responses to reviews
Reviewer #3
Abstract
Background: The justification of the study is lacking, and the purpose of the study is not highlighted. Please consider adding them.
The sentence was added as suggested Reviewer .
Lines 19-21. This sentence is not finished: "which indicates that a person weighing 60 kg and daily consuming 250 mg of EPA + DHA with mercury content in the muscles of eel (0.415 mg/kg) ….".
The sentence was changed as suggested Reviewer.
Line 22. This sentence, "The fatty acids indices (OFA, DFA, AI and TI) were also studied.", doesn't show any result.
The sentence was changed as suggested Reviewer.
Please delete (2), (3) and (4) from the abstract.
The (2), (3) and (4) was deleted as suggested Reviewer.
Introduction
Lines 64-66. Please consider rewriting this sentence "The metals belong to the group "trace elements", i.e. required in trace quantities (e.g. Fe, Zn, Cu and Mn) are also classified as heavy metals." for a better understanding.
The sentence was changed as suggested Reviewer.
Lines 66-68. Please consider rewriting this sentence "Besides these elements, heavy metals are also included chromium, cadmium, arsenic, lead, nickel, mercury and selenium." for a better understanding.
The sentence was changed as suggested Reviewer.
Lines 92-93. "According to these authors …". Please indicate which authors you are referring to.
The reference was added as suggested Reviewer.
Line 99-100. Please consider rewriting this sentence "Fish, especially fatty fish contain significant also contain the very important n-3 99 polyunsaturated fatty acids." for a better understanding.
The sentence was changed as suggested Reviewer.
Line 103. "According to these authors …". Please indicate which authors you are referring to.
The reference was added as suggested Reviewer.
Lines 113-114. Please indicate the references for the 4:1 and 10:1 ratios in the following sentence "The recommended ratio of n-6 to n-3 typically ranges from 4:1 (or less) to 10:1."
The reference was added as suggested Reviewer.
Lines 117-118. Please reformulate this sentence "In the literature, the nutritional quality indices are often found." for a better understanding.
The sentence was changed as suggested Reviewer.
Lines 118-121. Please indicate the references for this sentence "This may be related to the fact that all PUFAs in the diet can lower LDL-C, i.e. low-density lipoprotein cholesterol, and lower serum cholesterol, while all SFAs contribute to high serum cholesterol."
The reference was added as suggested Reviewer.
Lines 126-128. Please reformulate this sentence "Ulbricht and Southgate [40] and Chen et al. [41], as the inverse of the atherogenic index, named the HPI, which focuses on the effect of fatty acids composition on CVD." for a better understanding.
The sentence was changed as suggested Reviewer .
Lines 128-130. Please reformulate this sentence "Thrombogenicity index (TI) defined as the relationship between the pro-thrombogenic FAs (C12:0, C14:0, and C16:0) and the anti-thrombogenic FAs (MUFAs and the n-3 and n-6 PUFAs) [38,40]." for a better understanding.
The sentence was changed as suggested Reviewer .
Lines 136-137 show the experimental design, not the aim of the study. Also, the originality of the study is not clearly stated.
The aim of the study was changed as suggested Reviewer.
Materials and Methods
Lines 164-165. "Muscle tissue (without skin) from the dorsal part were prepared from one specimens". Was the muscle sample collected from only one specimen?
The sentence was changed as suggested Reviewer.

Round 2
Reviewer 3 Report
The authors have adequately addressed my comments in the revised version of the manuscript. Therefore, I have no further comments, and I recommend the acceptance of the manuscript in this form.